# METROGNN: METRO NETWORK EXPANSION WITH REINFORCEMENT LEARNING

## ABSTRACT

Selecting urban regions for metro network expansion that serve maximal transportation demands is critical to urban development, while computationally challenging to solve. First, metro network expansion is dependent on multiple complicated features, such as urban demographics, origin-destination (OD) flow, and relationships with existing metro lines, requiring a unified model to incorporate these correlated features for region selection. Second, it is a complex decision-making task with an enormous solution space and various constraints, due to the large number of candidate regions and restrictions on urban geography. In this paper, we present a reinforcement learning framework to solve a Markov decision process on an urban heterogeneous multi-graph, achieving metro network expansion by intelligently selecting a set of nodes on the graph. A novel graph neural network is proposed, which unifies the complicated features and learns effective representations for urban regions. In addition, we design an attentive reinforcement learning agent with action masks to efficiently search the large solution space and avoid infeasible solutions indicated by the various constraints. Experiments on real-world urban data of Beijing and Changsha show that our proposed approach can improve the satisfied transportation demands substantially by over 30% compared with state-of-the-art reinforcement learning methods. Further in-depth analysis demonstrates that MetroGNN can provide explainable results in scenarios with much more complicated initial conditions and expansion requirements, indicating its applicability in real-world metro network design tasks. Codes are released at https://anonymous.4open.science/r/MetroGNN-31DD.

## 1 INTRODUCTION

Metro network expansion is a geometrical combinatorial optimization (CO) problem, where a set of urban regions are selected for building metro stations from a large candidate region pool (Nikolić & Teodorović, 2013). The goal is to maximize the transportation demands between different regions that can be served by the expanded metro network, under a given budget. Meanwhile, there exist various constraints, such as station spacing and line straightness, restricting the feasible regions for selection. Due to its intrinsic complexity, it is almost impossible to obtain the optimal network expansion. Numerous computational approaches have been proposed to search for efficient metro network, however, their inferior performance indicates that this problem is still largely unsolved (Guihaire & Hao, 2008).

Solving metro network expansion is nontrivial due to two primary challenges. On the one hand, complicated features need to be taken into consideration. Specifically, numerical features capturing urban demographics of each region (Driscoll et al., 2018), matrix transportation flow between different regions, and the relationship with existing metro lines are all crucial factors for metro network expansion. It is necessary to model these features in diverse forms with unified representations. On the other hand, selecting regions for metro network expansion is an NP-hard problem with an enormous solution space composed of all candidate regions, which makes it impossible to conduct an exhaustive search (Darvariu et al., 2023; Yolcu & Póczos, 2019). For example, a medium-sized city with 1000 regions can have a solution space over $10^{30}$, far beyond what exact solution can handle. Meanwhile, the problem is even more complex due to the various constraints from urban

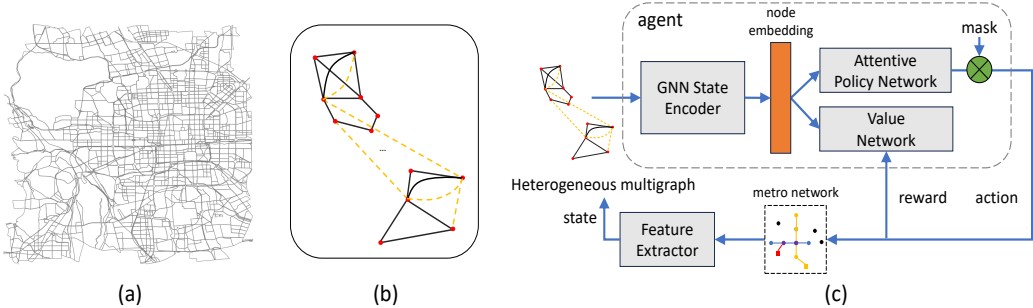

Figure 1: (a) Real region division in Beijing determined by road network. (b) Heterogeneous multigraph, where each red node represents a region. The black solid line and the orange dashed line are two types of edges, corresponding to spatial contiguity and OD associations between regions. (c) Schematic of our approach. At each step, the agent receives states and rewards from the environment and outputs the selected region for metro network expansion. Best viewed in color.

geography (Bagloee & Ceder, 2011), including the spacing and angles between stations and line segments.

Existing approaches for metro network expansion can be divided into two categories. First, heuristics are proposed to select regions, such as simple greedy rules(Zarrinmehr et al., 2016), or using genetic algorithms(Nayeem et al., 2018). Simulated annealing(Yu et al., 2012; Feng et al., 2018) and ant colony methods(Yu et al., 2005; Yang et al., 2007) are also adopted to better navigate the search space. However, the various constraints can not be well handled by these heuristics, which leads to infeasible solutions. Second, mathematical programming approaches(Wei et al., 2019; Gutiérrez-Jarpa et al., 2018) eliminate the solution space by restricting the selected regions in narrow corridors, and solve the reduced mixed integer programming problem using optimization solvers. Not surprisingly, the corridor approximation is oversimplified, which blocks out solutions of high quality and thus results in sub-optimal performance of the expanded metro network. Recently, Wei et al. (2020) made the first attempt to solve this problem via reinforcement learning (RL). Nonetheless, the correlated features are ignored by this approach, and the RL agent only considers the region location to build a new metro line. Moreover, they select regions on $n \times n$ grids, which induces large errors from the original problem form, where urban regions are actually a graph of irregularly-shaped geometries that distribute non-uniformly instead of simple grids.

In this paper, we propose a systematic RL framework for solving complex Markov decision process (MDP) on the graph. The metro network is expanded by selecting nodes intelligently on a heterogeneous multi-graph representing urban regions. To address the challenge of complicated features, we design a novel graph neural network (GNN) to learn effective representations for the heterogeneous multi-graph, which unify the complicated features with learnable graphical embeddings. Independent message propagation and neighbor aggregation are developed to capture both spatial contiguity and transportation flow between urban regions. To achieve efficient search of the solution space for NP-hard problem, we propose an attentive policy network with an action mask to select regions. The solution space is greatly explored by attending more on high-quality actions, and maintain necessary exploration of regions with low benefits. Meanwhile, with the various constraints of metro network well handled by the action mask, we guarantee the feasibility of the obtained solutions.

To summarize, the contributions of this paper are as follows,

- We propose a graph-based RL framework for solving complex MDP, which is able to address the challenging geometrical CO problem of metro network expansion.

- We design a novel GNN and an attentive policy network with an action mask to learn representations for urban regions and select new metro stations. The proposed model encodes complicated features into unified graphical representations, and successfully search the large solution space.

- Extensive experiments are conducted on metro network expansion using real-world urban data of Beijing and Changsha, and the results demonstrate that the proposed MetroGNN can substantially improve OD flow satisfaction by over 30% against state-of-the-art approaches.

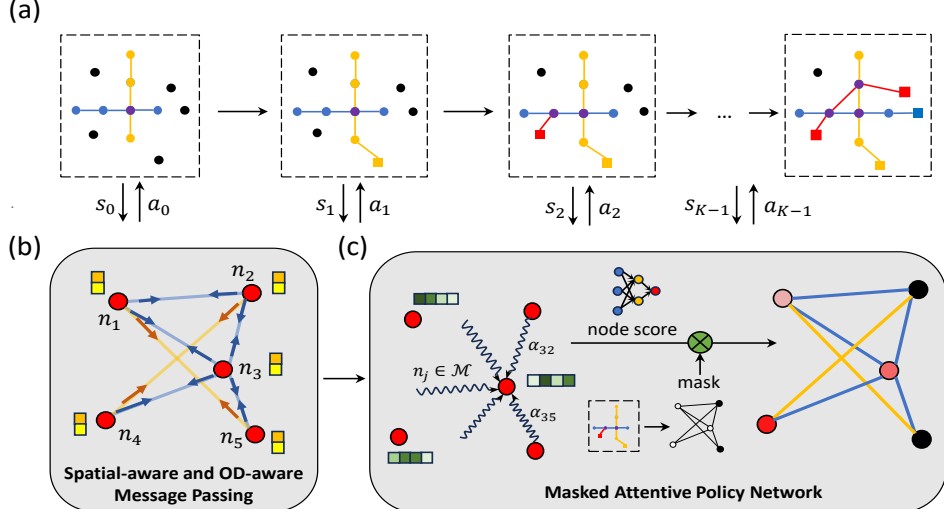

Figure 2: (a) Metro network expansion. At each step, the agent selects a node that either extends existing lines ($a_0$ for the yellow line) or constructs new lines ($a_1$ for the red line). We use distinct colors for various lines, and highlight interchange points in purple. (b) The proposed GNN model. We design spatial-aware and OD-aware message passing to learn effective node representations. (c) The proposed masked attentive policy network for node selection. Infeasible regions are blocked by the action mask, and an attention module is designed to capture the relationship between each node and the metro network. Best viewed in color.

## 2 PROBLEM STATEMENT

Given a set of nodes, $\mathcal{N} = \{n_1, n_2, ..., n_k\}$, representing the centroids of urban regions that are divided by the road network (see Figure 1(a)), a metro network $\mathcal{M} = (\mathcal{V}, \mathcal{E})$ can be described with a subset of nodes $\mathcal{V} \in \mathcal{N}$, and the edges $\mathcal{E}$ (metro lines segments) connecting nodes. Then metro network expansion can be defined as the sequential selection of nodes for station construction that grows a metro network to maximize its total satisfied OD flow, quantified as follows:

$$C_{od}(\mathcal{M}) = \sum_{(n_i, n_j) \in \mathcal{E}} \frac{\text{EucDis}(n_i, n_j)}{\text{PathDis}(n_i, n_j)} * \mathcal{F}_{ij}, \tag{1}$$

where $\text{EucDis}(n_i, n_j)$ is the distance between $n_i$ and $n_j$, $\text{PathDis}(n_i, n_j)$ is the path length between $n_i$ and $n_j$ via $\mathcal{M}$, and $\mathcal{F}_{ij}$ denotes the OD flow between $n_i$ and $n_j$.

Particularly, metro lines must adhere to constraints such as total budget, spacing between stations, line straightness, *etc*. The problem can be formulated as follows:

**Input**: The metro network $\mathcal{M} = (\mathcal{V}, \mathcal{E})$ with an OD flow matrix $\mathcal{F}$, construction cost $C$, total budget $B$, and maximum number of new metro lines constructed $L$.

**Output**: A sequence of regions $S$ from $\mathcal{N}$ representing the expansion order of stations.

**Objective**: Maximize $C_{od}(\mathcal{M})$, the overall satisfied OD of the expanded metro network.

## 3 METHOD

### 3.1 OVERALL FRAMEWORK

We propose a graph-based RL framework to solve the complex MDP, where the agent selects one node at each step and connects it to the current metro network, on a heterogeneous multi-graph (Figure 2(a)). Specifically, to incorporate complicated features, we design a novel GNN state encoder to learn effective representations for urban regions, unifying these features into graphical embeddings. As shown in Figure 2(b), independent message passing mechanisms for spatial contiguity and OD flow are developed to capture different information through the heterogeneous edges. To efficiently search the enormous solution space, we propose an attentive policy network with a carefully designed action mask, as illustrated in Figure 2(c).

## 3.2 MARKOV DECISION PROCESS

We propose a DRL model to solve the sequential decision-making problem, where an intelligent agent learns to automatically select regions for expansion by interacting with the metro network environment, as shown in Figure 1(c). The metro network expansion can be expressed as a Markov Decision Process (MDP) with the following critical components:

- **State.** The state $S_t$ is a three-tuple $(\mathcal{M}_t, b_t, l_t)$ containing the current metro network $\mathcal{M}_t = (\mathcal{V}_t, \mathcal{E}_t)$, the remaining budger $b_t$, and the number of new metro lines that can be built $l_t$.

- **Action.** The action $A_t$ corresponds to the selection of a single node in $\mathcal{N}$. Considering that the metro network expansion includes the extension of existing lines and the construction of new lines and needs to satisfy corresponding constraints, available actions are defined as:

$$A(S_t = (\mathcal{M}_t, b_t)) = \{n \in \mathcal{N} \mid n \in ((\mathcal{X}^e(\mathcal{M}_t, b_t) \cup (\mathcal{X}^c(\mathcal{M}_t, b_t, l_t)),$$

where $\mathcal{X}^e(\mathcal{M}_t, b_t)$ and $\mathcal{X}^c(\mathcal{M}_t, b_t, l_t)$ represents to the set of regions that are available for extension and construction based on specific constraints (definition in Section A.1), respectively.

- **Reward.** Referring to the definition of total satisfied OD flow in Equation 1, the intermediate reward $R_t$ for action $A_t$ is defined as $C_{od}(\mathcal{M}_t) - C_{od}(\mathcal{M}_{t-1})$.

## 3.3 HETEROGENEOUS MULTI-GRAPH MODEL

We utilize a heterogeneous multi-graph to faithfully describe the urban regions. In this graph model, the node set $\mathcal{N} = \{n_1, \cdots, n_k\}$ represents the regions divided by the road network. We introduce two types of edges to effectively capture the relationships between regions, as illustrated in Figure 1(b). The first type links contiguous nodes, capturing their proximity on small spatial scales. The second type connects pairs of nodes with significant OD trips, capturing flow patterns between urban regions on a larger scale. In specific, the heterogeneous edges are denoted as follows,

$$e_{ij}^s = \mathbb{1}\{0 < \text{EucDis}(n_i, n_j) \leq t_1\}, \quad e_{ij}^o = \mathbb{1}\{\mathcal{F}_{ij} \geq t_2\}, \quad \forall n_i, n_j \in \mathcal{N} \tag{2}$$

where $t_1$ and $t_2$ are threshold values.

We conduct preprocessing on this graph to facilitate reasonable metro network expansion (see Section A.2). By expressing the problem with a graphical model, our framework can comprehensively express the spatial relationships and OD characteristics within the city.

## 3.4 ENCODING COMPLICATED FEATURES WITH GNN

We design a novel GNN model as the encoder to learn unified representations of complicated features for regions through the heterogeneous edges. Two groups of features are incorporated for each region. The first group directly relates to the OD trips of the region, which includes the total OD access flows $F_1^D$, the OD flows with neighboring regions $F_2^D$ and the OD flows with the regions $\mathcal{V}$ where metro stations located $F_3^D$. And the second group contains auxiliary features, including the population size $F_1^A$, the type and number of Points of Interests (POIs) in each urban region $F_2^A$, as well as topological features in the graphical model (including $F_4^A, F_5^A$ and $F_6^A$), such as degree.

In order to obtain a unified representation of these complicated features, we first encode them into dense embeddings for each node with the input attributes $\mathbf{A}_i = [F_1^A \parallel \ldots \parallel F_3^A \parallel F_1^D \parallel \ldots \parallel F_6^D]$. Then we design independent message propagation and neighbor aggregation for the heterogeneous graph, capturing spatial contiguity and OD flow, respectively. We let the set $\mathcal{X}_i^s$ and $\mathcal{X}_i^o$ consist of nodes connected to $n_i$ via $e_{ij}^s$ and $e_{ij}^o$, respectively. As demonstrated in Figure 2(b), we aggregate neighbors' representations at different scales through two types of heterogeneous edges, as follows,

$$\mathbf{h}_{s,i}^{(l)} = \sum_{n_j \in \mathcal{X}_i^s} \mathbf{W}_s^{(l)} \mathbf{h}_j^{(l)}, \quad \mathbf{h}_{o,i}^{(l)} = \sum_{n_j \in \mathcal{X}_i^o} \mathbf{W}_o^{(l)} \mathbf{h}_j^{(l)}, \quad \mathbf{h}_i^{(0)} = \mathbf{W}_A \mathbf{A}_i \tag{3}$$

where $\mathbf{W}_A$, $\mathbf{W}_s$ and $\mathbf{W}_o$ are linear transformation layer, $\mathbf{h}_i^{(0)}$ is the dense embedding obtained from the initial encoding. Next, we update the node embeddings by combining neighbor information and the node itself, as follows,

$$\mathbf{h}_i^{(l+1)} = \tanh(\mathbf{W}_c^{(l)}(\mathbf{h}_{s,i}^{(l)} \parallel \mathbf{h}_{o,i}^{(l)}) + \mathbf{h}_i^{(l)}), \tag{4}$$

where $\mathbf{W}_c^{(l)}$ is a linear transformation layer, $\|$ means concatenation.

With the proposed GNN model, we unify the complicated features for metro network expansion, and obtain effective node representations from these features, with spatial contiguity and OD flow information injected into them. The representations from GNN are shared across the policy and value networks, enabling effective node selection and return prediction.

## 3.5 PLANNING WITH MASKED ATTENTIVE POLICY NETWORK

As a conservative estimate, assume that each region is spatially adjacent to 4 other regions, and the metro network initially has 3 lines, with 40 more stations to be added. This results in a total of approximately $C_{40}^5 * 4^{40} \approx 10^{30}$ possible solutions. The solution space expands exponentially with the number of expansion stations, making it exceedingly challenging to find optimal solutions, especially when considering various constraints such as straightness and spacing. To search the massive solution space under various constraints, we propose an attentive policy network with an action mask for efficient exploration of feasible solutions. The proposed policy network assigns scores to feasible nodes based on embeddings computed by GNN, substantially reducing the action space and emphasizing high-quality node selection.

**Attentive policy and value network.** To effectively explore the large solution space, we utilize attention to emphasize actions of high quality that are strongly correlated with the current metro network. Specifically, as shown in Figure 2(c), we measure the relevance to the current metro network of each node $n_i$ using representations from GNN as follows,

$$w_{ij} = \frac{(\mathbf{W}_Q \mathbf{h}_i^{(L)})^T (\mathbf{W}_K \mathbf{h}_j^{(L)})}{\sqrt{d}}, \quad \alpha_{ij} = \frac{\exp(w_{ij})}{\sum_j \exp(w_{ij})}, \quad \forall n_i \in \mathcal{N}, v_j \in \mathcal{V}, \quad (5)$$

where $d$ is the embedding dimension, $\mathbf{W}_Q, \mathbf{W}_K$ are learnable parameters, and $\alpha_{ij}$ is the attention score. Node scores are then computed according to the attentive importance through a multi-layer perceptron (MLP) as follows,

$$\mathbf{a}_i = \tanh\left(\sum_{v_j \in \mathcal{M}} \alpha_{ij} \cdot \mathbf{h}_j^{(L)} + \mathbf{h}_i^{(L)}\right), \quad s_i = \mathtt{MLP}_p(\mathbf{a}_i), \quad p(n_i|\mathcal{M}) = \frac{\exp(s_i)}{\sum_i \exp(s_i)} \quad (6)$$

where nodes are selected and added to the metro network based on a probability distribution determined by the score $s_i$ with carefully-designed action mask (see Section A.1).

A value network is also developed, which takes the average of node embeddings to summarize the current network state and uses an MLP to estimate returns as follows,

$$\mathbf{a}_{avg} = \frac{1}{|\mathcal{N}|} \sum_{i=1}^{|\mathcal{N}|} \mathbf{a}_i, \quad \hat{r} = \mathtt{MLP}_v(\mathbf{a}_{avg}), \quad (7)$$

where $r$ is the estimated performance of metro network expansion.

By training the proposed model with Proximal Policy Optimization (PPO) (Schulman et al., 2017), our proposed approach can effectively avoid infeasible solutions via action mask, and at the same time, efficiently explore the vast solution space, since most nodes of low quality in the original action space will not be frequently sampled by the proposed attentive policy and value network.

## 4 EXPERIMENTS

### 4.1 EXPERIMENT SETTINGS

**Data.**

We conduct experiments based on the metro networks in two of the largest cities in China, Beijing and Changsha. Specifically, we adopt the actual urban region divisions split by the authentic road structure. Real OD flow data for the whole year of 2020 is utilized, which is collected from Tencent Map, a prominent mapping and transportation service application in China. Table 1 shows the basic information of the dataset.

**Baselines and evaluation.** We compare our model with mathematical programming approaches (Wei et al., 2019) utilizing two different solvers, CBC (MPC) and GUROBI (MPG). Heuristics baselines are also compared, including Greedy Strategy (GS) (Laporte & Pascoal, 2015), Genetic Algorithm (GA) (Owais & Osman, 2018), Simulated Annealing Algorithm (SA) (Fan & Machemehl, 2006), and Ant Colony Optimization (ACO) (Yang et al., 2007). We further include the state-of-the-art RL approach, DRL-CNN (Wei et al., 2020) for comparison. Details of the adopted baselines are provided in Section A.3. For each method, we vary the seeds and conduct each experimental configuration 10 times. To evaluate the effect of metro network expansion, we calculate the OD flows satisfied by the expanded metro network according to (1).

**Model Implementation.** We implement the proposed model using PyTorch (Paszke et al., 2019). We carefully tune the hyper-parameters of our model, including the learning rate, regularization, *etc* (details in Section A.4). The model is trained on a single server equipped with an Nvidia GeForce 2080Ti GPU, which typically costs approximately 8 hours.

| City | Beijing | Changsha |
|------|---------|----------|
| Nodes | 1166 | 469 |
| Edges | 4656 | 1642 |
| Avg Degree | 3.99 | 3.50 |
| Avg Area | $1.21km^2$ | $1.17km^2$ |
| T-APL | 20.37 | 13.02 |
| T-MSPL | 55 | 39 |
| M-APL | 34.38km | 20.81km |
| M-MSPL | 101.73km | 63.48km |

Table 1: Statistical Overview of the Dataset. **T** denotes topological measures, while **M** denotes metric measures. **APL** represents average path length, and **MSPL** refers to maximum shortest path length. For example, T-APL represents the average topological path length.

## 4.2 PERFORMANCE COMPARISON

We evaluate each method under different scenarios by changing the total budget for expansion in billion to various values and assessing the corresponding performance. Results of our model and baselines are illustrated in Table 2. From the results, we have the following observations,

- **Heuristic algorithms are ineffective for metro network expansion.** Among all the methods, heuristic algorithms are the least effective, with the satisfied OD only about 50% of other approaches in most cases. Particularly, they often generate infeasible solutions in the initial population, leading to insufficient exploration of the solution space.

- **DRL-based methods have significant advantages over other approaches.** DRL-CNN outperforms other baselines in most cases, achieving higher satisfied OD with an average improvement of 4.4%, demonstrating the superior ability of RL to search a large solution space. Nevertheless, DRL-CNN suffers from severe performance deterioration in complicated scenarios (B=60), with the satisfied OD 5.1% worse than MPC and MPG on average.

- **Our proposed model achieves the best performance in different scenarios.** Our approach substantially surpasses existing baselines under all budgets, substantially improving the satisfied OD flow by over 15.9% against the best baseline in average of three different expansion budgets. Notably, in contrast to DRL-CNN that fails to outperform baselines in complicated scenarios, our approach exhibits more significant advantages in complicated scenarios with a higher budget, with improvements on satisfied OD even over 30%. The proposed novel GNN that unifies complicated features is able to capture the intricate connections between regions and the underlying transportation patterns, which guarantees decent performance and avoids the oversimplification of grid-based approximation adopted by DRL-CNN.

## 4.3 METRO NETWORK EXPANSION IN COMPLICATED SCENARIOS

To further explore the performance in various scenarios, we evaluate different methods by varying the number of initial lines (IL) as well as the maximum number of newly constructed lines (ML), as shown in Figure4. Notably, our method exhibits significant advantages in diverse scenarios, and the **advantages become more pronounced as the complexity of the scenarios increases**. For instance, when expansion permits the construction of more lines, the improvement against DRL-CNN increases from 6.8% to an impressive 23.3%. The improvement also gradually increases from a minimum of 16.4% to 27.6% as the complexity of the initial metro network rises.

| Method | Beijing | | | Changsha | | |
|---|---|---|---|---|---|---|
| | B=40 | B=50 | B=60 | B=40 | B=50 | B=60 |
| GS | $8.25_{\pm0.00}$ | $9.31_{\pm0.00}$ | $10.40_{\pm0.00}$ | $10.11_{\pm0.00}$ | $11.26_{\pm0.00}$ | $12.58_{\pm0.00}$ |
| GA | $9.95_{\pm1.78}$ | $10.13_{\pm1.98}$ | $12.87_{\pm2.27}$ | $14.24_{\pm1.46}$ | $15.34_{\pm1.65}$ | $16.55_{\pm1.89}$ |
| SA | $9.59_{\pm1.57}$ | $10.70_{\pm1.59}$ | $12.29_{\pm2.08}$ | $13.84_{\pm1.63}$ | $15.02_{\pm1.55}$ | $16.39_{\pm1.72}$ |
| ACO | $11.01_{\pm1.14}$ | $12.42_{\pm1.30}$ | $13.66_{\pm1.45}$ | $16.61_{\pm1.59}$ | $17.67_{\pm1.52}$ | $17.17_{\pm1.97}$ |
| MPC | $14.40_{\pm0.28}$ | $15.11_{\pm0.61}$ | $16.60_{\pm1.19}$ | $17.47_{\pm0.60}$ | $18.34_{\pm0.91}$ | $\underline{20.43}_{\pm1.30}$ |
| MPG | $14.40_{\pm0.28}$ | $15.16_{\pm0.93}$ | $\underline{16.81}_{\pm1.13}$ | $17.47_{\pm0.91}$ | $18.07_{\pm1.26}$ | $20.17_{\pm1.63}$ |
| DRL-CNN | $\underline{14.46}_{\pm0.92}$ | $\underline{15.78}_{\pm1.33}$ | $16.38_{\pm1.48}$ | $\underline{18.30}_{\pm1.03}$ | $\underline{18.98}_{\pm1.98}$ | $19.21_{\pm1.81}$ |
| MetroGNN (ours) | $\mathbf{15.88}^{*}_{\pm0.73}$ | $\mathbf{18.93}^{**}_{\pm0.87}$ | $\mathbf{21.45}^{**}_{\pm1.02}$ | $\mathbf{20.79}^{*}_{\pm1.07}$ | $\mathbf{22.72}^{**}_{\pm1.13}$ | $\mathbf{24.65}^{**}_{\pm1.36}$ |
| impr% v.s. DRL-CNN | +9.8% | +20.0% | +31.0% | +13.6% | +19.7% | +28.3% |

Table 2: Evaluation of metro network expansion performance across varying budgets (**B**), higher is better. Statistical significance is determined using a t-test to compare MetroGNN with DRL-CNN, denoted as $^{*}$p-value $< 0.1$ and $^{**}$p-value $< 0.05$.

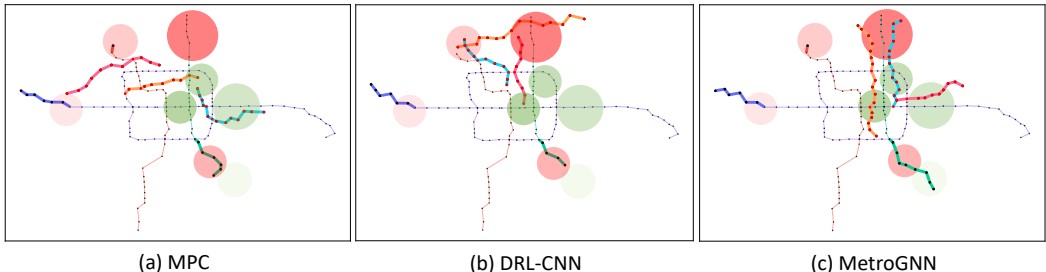

(a) MPC            (b) DRL-CNN            (c) MetroGNN

Figure 3: Metro network expansion results by MPC, DRL-CNN and MetroGNN for Beijing. We use colors to distinguish between different metro lines, and use boldface to indicate expansion outcome from initial conditions. Black dots indicate stations on the initial lines and their extensions, and red dots represent stations on new lines. Red and green circular indicate areas where population and POIs are clustered, respectively. The darker the color, the higher the density. Best viewed in color.

To provide a deeper understanding of the reliability and practical applicability of our planning solution within real-world contexts, we offer detailed explanations for the outcomes of our model. In Figure 3, we present the results generated by the MP, DRL-CNN, and MetroGNN methods based on Beijing, respectively. Specifically, we illustrate the areas where populations and POIs are densely clustered to analyze the generated results. Compared to baselines, the planning solution generated by our approach covers almost all the areas with high population and POI densities, which naturally correspond to numerous travel demands.

Notably, our solution takes into account the efficiency of the transportation network. As shown in Figure 3, the new lines generated by our approach are interconnected, and each new line introduces at least two additional interchange stations to the metro network. These comprehensive analyses demonstrate the practical implications and strengths of the proposed MetroGNN expansion strategy.

## 4.4 ABLATION STUDY

We conduct ablation experiments to showcase the efficacy of the graph model and the incorporated complicated features, and the results are consolidated and presented in Figure 5.

**Graph Modeling.** The urban regions exhibit intricate spatial correlations characterized by both short-range proximity and long-range OD flow patterns. By harnessing the graph modeling approach and GNN, our approach effectively captures these complexities among urban regions. As shown in Figure 5(a), when the graph model is omitted, the satisfied OD flow of the expanded metro network drops significantly from 21.80 to only 14.02. This stark change underscores the pivotal role of the graph model in the metro network expansion task.

**Spatial-aware and OD-aware Message Passing.** In the proposed GNN model, we design two independent message propagation mechanisms, spatial-aware and OD-aware message passing. We experiment on two variants of our GNN model, each preventing message propagation through one kind of edge, to study their corresponding effect. As illustrated in Figure 5(a), removing either

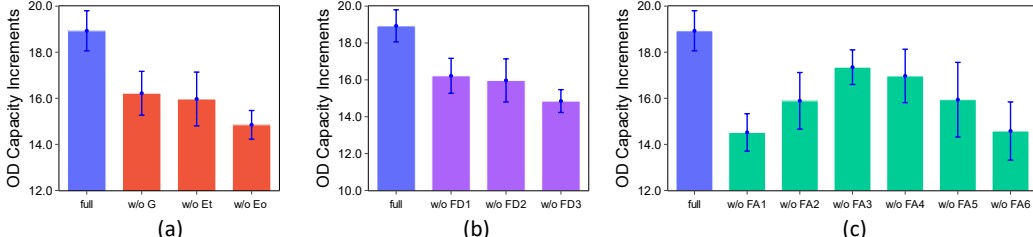

Figure 4: Performance of MPC, DRL-CNN and MetroGNN methods for different IL and ML for Beijing (BJ) and Changsha (CSX), respectively, higher is better. Best viewed in color.

Figure 5: Performance of MetroGNN and its variants that remove different elements, including whole graph model (G), spatial edges (Et), transportation flow edges (Eo), OD direct (FD) and auxiliary (FA) features. Best viewed in color.

spatial or transportation edges leads to a significant deterioration in performance, with a decrease of 28.8% and 30.8%, respectively. These results affirm the essential roles of spatial-aware and OD-aware message propagation, which aggregates a wide range of OD flow information to nodes, facilitating our proposed GNN in acquiring meaningful node representations.

**OD Direct and Auxiliary Features.** As shown in Figure 5(b), when the three OD direct features are excluded, our method observes varying degrees of performance degradation. In particular, removing FD3 results in the largest performance drop (-20.4%), which is reasonable since it reflects the direct benefit of adding a region to the metro network.

We also add auxiliary features to support decent metro network expansion, since they are valuable supplementary information associated with OD trips. As demonstrated in Figure 5(c), removing these auxiliary features indeed leads to worse performance, with the satisfied OD flow dropping by 8.4% to 19.9%. Among these features, removing population (FA1) brings about the largest deterioration, as population information is quite important when considering metro network expansion. Meanwhile, terminal station (FA6) also plays a vital role, and removing it leads to a 22.8% drop in satisfied OD flows.

## 4.5 CONVERGENCE ANALYSIS

Reducing the action space is essential for enhancing the efficiency of model training, especially for metro network expansion that exhibits an enormous search space. Here, we remove the attention module from the policy network and only retain the MLP module, to investigate the impact of the attentive policy network on the solution space. As illustrated in Figure 6(a), when employing the attentive policy network, our approach achieves performance close to that of DRL-CNN (14.81) and converges after only about 25 iterations. In contrast, the performance of the MLP-only policy network is only comparable to the worst baseline (9.60) at this point, and does not converge until approximately 50 iterations. As shown in Figure 6(b), where we show the probability distribution of the first action, it is evident that the probabilities of the attentive policy network are almost concentrated in one region, while the MLP-only policy network exhibits a more even probability distribution across four different regions. This observation highlights the significance of the attention mechanism in the policy network. It guides the agent to focus more on high-quality nodes with a strong correlation to the current network, promoting more efficient exploration.

We also investigate the influence of critical hyper-parameters and the transferability of the proposed model, which are provided in Section A.6 and Section A.5, respectively. Section A.7 shows how our method responds to different evaluation metrics, such as social equity.

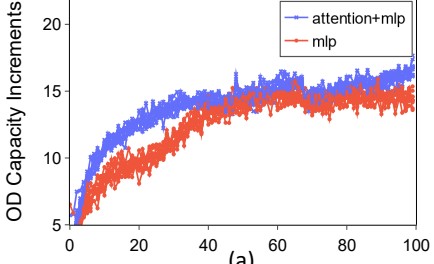 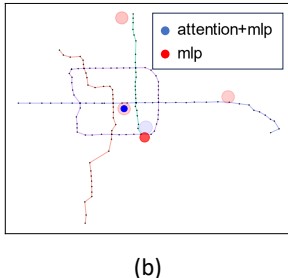

Figure 6: (a) Schematic of the probability distribution (greater than 1e-3) of the first action after 40 iterations of training. Each circle corresponds to a candidate region, and darker color and smaller radius indicate higher selection probabilities. (b) The effect of the attention design in policy network on model training convergence. Best viewed in color.

## 5 RELATED WORK

**Deep Reinforcement Learning for Planning.** Deep reinforcement learning (DRL) (Mnih et al., 2015) approaches, which combines RL algorithms with deep neural networks (DNN) as function approximators, have demonstrated excellent performance and become the new state-of-the-art, especially in complex planning tasks such as gaming (Mnih et al., 2013; Silver et al., 2016; Ye et al., 2020), autonomous driving (Kiran et al., 2021; Li et al., 2022; Shi et al., 2020), natural language processing (Li et al., 2016; He et al., 2015; Tan et al., 2022), recommender system (Zheng et al., 2018; Zhao et al., 2018; Wang et al., 2021b), and solving mathematical problems Fawzi et al. (2022); Bello et al. (2016); Bengio et al. (2021). On the one hand, the efficient approximators such as DQN (Mnih et al., 2013; 2015) can accurately handle high-dimensional inputs. On the other hand, the actor-critic framework (Konda & Tsitsiklis, 1999) is effective in narrowing down the action space and improving exploration efficiency.

**Transportation Network Design.** As a widely studied engineering task (Guihaire & Hao, 2008; Ibarra-Rojas et al., 2015; Farahani et al., 2013; Kepaptsoglou & Karlaftis, 2009; Laporte et al., 2011; Wang et al., 2021a), existing solutions to this problem can be classified into three primary categories: mathematical programming methods, heuristic methods, and deep reinforcement learning methods. Mathematical programming methods(Escudero & Muñoz, 2009; Wang et al., 2023; Gutiérrez-Jarpa et al., 2013; Owais et al., 2021) typically formulate it as an Integer Linear Programming (ILP) problem and then apply a solver to find the optimal solution. For example, Wei et al. (2019) applied the minimum distance to the utopia point to handle the bi-objective Mixed-Integer Linear Programming (MILP) problem. However, the computational complexity increases drastically as the problem scales up, making it challenging to find optimal solutions within a reasonable time limit. Heuristic methods, including Tabu Search (Dufourd et al., 1996; Fan & Machemehl, 2008), Simulated Annealing (Fan & Machemehl, 2006; Kumar et al., 2020; Zhao & Zeng, 2006), and Evolutionary Algorithms (Mumford, 2013; Arbex & da Cunha, 2015; Chakroborty, 2003; Nayeem et al., 2018; Bourbonnais et al., 2021), have also emerged to address this problem. However, it is difficult to handle the diverse constraints of metro networks with suitable heuristic operators, and these methods often invest substantial time in exploring impractical solutions. Recently, Wei et al. (2020) explored the usage of RL for metro line planning. However, they ignore the complicated features, and the adopted grid-like city representation induces significant errors from real-world scenarios.

## 6 CONCLUSION

In this paper, we investigate the problem of metro network expansion, and propose MetroGNN, a systematic graph-based RL framework that can solve complex node selection MDPs on the graph. The proposed model unifies complicated features with GNN and explores the solution space efficiently with an attentive policy network and a carefully designed action mask. Through extensive experiments, we demonstrate the effectiveness of our approach, which can improve the satisfied OD flow by over 15.9% compared to state-of-the-art baselines. Notably, the advantage of our method is even more significant in complex scenarios, indicating its potential in solving real-world metro network design problems where there tend to be much more complicated initial conditions and constraints. Looking ahead, we plan to investigate the performance of the proposed systematic RL framework in other graph-based decision tasks, such as influence maximization on a social network.

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

# A  APPENDIX

## A.1  RULES FOR METRO NETWORK EXPANSION

The expansion of the metro network is subject to specific constraints to ensure the systematic and feasibility. To fully describe these constraints, we introduce the following notations: $\mathcal{V}\pm 1 \subset \mathcal{V}$ denotes the set of terminal stations for all metro lines, and $\mathcal{V}\pm 2 \subset \mathcal{V}$ denotes the set of subterminal stations directly connected to the terminal stations in $\mathcal{V}_{\pm 1}$. As introduced in Section 3.2, the expansion of metro network can be divided into the extension of existing lines and the construction of new lines. Regions available for the extension of existing lines must maintain reasonable station spacing and avoid sharp bends in the topology, as follows,

$$t^e(\mathcal{M}) = \{n_i \in \mathcal{N}\backslash\mathcal{V} \,|\, \mathrm{EucDis}(n_i, n_j) \in [t3, t4] \wedge \angle n_i n_j n_k \in [90°, 180°] \wedge n_j \in \mathcal{V}_{\pm 1} \wedge n_k \in \mathcal{V}_{\pm 2} \wedge (n_k, n_j) \in \mathcal{E}\}$$

where $t^e(\mathcal{M})$ represents the set of topologically feasible regions for extension, and $t_3$, $t_4$ are predefined thresholds.

Taking into account the budget constraint, we use $\mathcal{C}^e(n, \mathcal{M})$ to denote the minimum cost of adding region $n$ in an extended manner, then the set of the feasible regions for extension $\mathcal{X}^e(\mathcal{M}, b_t)$ can be expressed as follows,

$$\mathcal{X}^e(\mathcal{M}, b_t) = \{n \in t^e(\mathcal{M}) \,|\, |\mathcal{C}^e(n, \mathcal{M})| \le b_t\}.$$

If a region is unsuitable for extension, we then consider whether it can serve as the starting station for a new metro line, topologically governed only by spacing rules,

$$t^c(\mathcal{M}) = \{n_i \in \mathcal{N}\backslash\mathcal{V} \,|\, \mathrm{EucDis}(n_i, n_j) \in [t3, t4]\}.$$

When considering limitations on the number of lines and budgets, the feasible regions for constructing new lines can be formulated as

$$\mathcal{X}^c(\mathcal{M}, b_t, l_t) = \{n \in t^c(\mathcal{M}) \,|\, |\mathcal{C}^c(n, \mathcal{M})| \le b_t \wedge l_t > 0\},$$

where $\mathcal{C}^c(n, \mathcal{M})$ denotes to the minimum cost of adding region $n$ in an constructive manner.

Despite the feasibility of regions, prioritization exists in the way regions expand the metro network. Specifically, if region $n_i$ satisfies both $n_i \in \mathcal{X}^e(\mathcal{M}, b_t)$ and $n_i \in \mathcal{X}^c(\mathcal{M}, b_t, l_t)$ simultaneously, it will be prioritized as an extension of an existing line. If $n_i$ can be used as an extension of multiple line, the least costly extension is considered.

## A.2  PREPROCESS OF THE GRAPH AND OD FLOWS

**Graph model.** To facilitate proper metro network expansion, the delineation of regions follows specific preprocessing steps. Initially, following the methodology outlined in Wei et al. (2020), smaller regions that are geographically close to each other are merged. This approach prevents excessive delineation of the road network structure. Additionally, regions with larger areas but lower traffic demands, typically corresponding to remote areas such as mountains or rivers, are excluded. This step focuses the analysis to regions characterized by higher travel demand and activity. Furthermore, regions that are distant from the initial metro lines are also omitted from consideration. This is often due to budgetary constraints, as these remote regions are challenging to access within the constraints of the available budget. Through these preprocessing steps, we can obtain a series of appropriate regions, laying the foundation for an efficient and fast strategy for metro network expansion.

**OD flows.** We obtain inter-regional OD flows by processing a large number of user spatio-temporal trajectories. Specifically, if an individual travel from region A to region B and remains in B for at least 15 minutes, this travel event contributes a count of 1 to the OD flows from A to B. Furthermore, if the individual subsequently travels to region C, the entire movement will be considered as two separate trips: one from A to B and another from B to C.

A.3 IMPLEMENTATION DETAILS OF BASELINE

We compare our MetroGNN method with the following baselines.

- **GS** selects new region that meets the largest OD trips with previously selected region at each step.

- **GA** generates an initial population of metro lines and employs well-designed crossover and variance operators on individuals to generate the solutions. We set the initial population size as 200 and limit the number of iteration to 2000. We also designed genetic operators for crossover and mutation which ensure the viability of the new population, and the probability of crossover and mutation are both 0.8.

- **SA** commences with an initial solution and introduces stochastic modifications to explore the solution space. The algorithm progressively adapts to embrace the stochastic nature of suboptimal solutions during iterations, enabling it to transcend local optima and advance towards improved solutions. We set the initial temperature to 1500, the cooling coefficient to 0.98, the termination temperature to 0.1, and 200 iterations for each temperature. Additionally, we set the acceptance threshold of SA to 0.1.

- **ACO** runs with agents deposit pheromones on paths and establish connections with probabilistic rules. It iteratively updates pheromone levels based on evaluation metrics, thus steering the agents to better solutions. We limit the maximum number of iterations of ACO to 3000 and include 128 instances per iteration. Other parameters of ACO aligns with Yang et al. (2007).

- **MP** formulates the metro expansion problem as an mixed integer programming model, and using solver to obtain solution. Considering the solution time, we use part of the expansion solutions of DRL-CNN and MetroGNN as a reference for the corridors to reduce to solution space. Specifically, we consider regions within 10km of the existing metro network as the corridors, where regions are available candidates for metro network expansion. Additionally, the final metro network generated by MP may be discrete and need be manual adjustment.

- **DRL-CNN** trains a actor-critic model to design new metro line based on the hidden state of current metro network. At each step, it takes the embedded features as reference and the hidden state as query to select next region. We adopt the network structure from Wei et al. (2020) to generate the metro network expansion solutions.

A.4 EXPERIMENT SETTINGS

For the metro network expansion task, there are four key parameters, construction cost, construction budget, initial metro lines (IL) and maximum new lines (ML). We list all the values of above parameters for metro network expansion and the hyper-parameters of our method in Table 3, with default parameters indicated in bold. Specifically, for the estimation of construction cost, we adopted the values of Wang et al. (2023) and fixed this setting in subsequent experiments.

A.5 THE TRANSFERABILITY.

In this section, we demonstrate how our proposed method adapts to the dynamics of growing cities and changing OD flows. To simulate the expansion of the city, we initially remove the outer 20% regions of Beijing and train MetroGNN on this reduced city for metro network expansion. Subsequently, we apply the model to directly generate expansion solutions for the complete city area, and we also fine-tune the model for 10 additional iterations and evaluate its performance. As shown in the Table 4, the trained model can directly provide new metro network expansion solutions for the expanded city with performance close to that of DRL-CNN. Remarkably, with further fine-tuning on the expanded city, the model produces outstanding expansion solutions, showcasing a significant improvement of more than 10.2% compared to DRL-CNN. Furthermore, recognizing that OD flows may change as urban functions shift, we explore the performance of the proposed method when confronted with varying OD trips. As illustrated in Table 5, in the face of unpredictable OD trips, MetroGNN provides expansion solutions with an improvement of over 12.84% compared to DRL-CNN, and the p-value $< 0.3\%$ further confirms the statistical significance of this substantial improvement. Notably, the well-trained model can provide effective solutions for different urban areas within 20 seconds. In contrast, other methods either yield solutions of poor quality or need to

Table 3: Parameter values of MetroGNN.

| Category | Parameter | Value |
|---|---|---|
| **Cost** | Cost per normal station (million RMB) | 300 |
| | Cost per interchange station (million RMB) | 600 |
| | Cost per kilometer (million RMB) | 500 |
| **Expansion** | The number of Initial lines | 2,**4**,6 |
| | Maximum new lines | 2,**3**,4 |
| | Budget for expansion (billion RMB) | 40,**50**,60 |
| **Network** | GNN layer | 2 |
| | GNN node dimension | 32 |
| | Attention Head | 2 |
| | Policy Head `MLP`$_p$ | [32, 1] |
| | Value Head `MLP`$_v$ | [32, 32, 1] |
| **PPO** | gamma | 0.99 |
| | tau | 0 |
| | Entropy Loss $\beta$ | 0.01 |
| | Value Loss $\gamma$ | 0.5 |
| **Train** | optimizer | Adam |
| | weight decay | 0 |
| | learning rate | 0.0004 |

Table 4: The transferability of MetroGNN on expended city compared with DRL-CNN. Statistical significance is determined using a t-test to compare MetroGNN with DRL-CNN, denoted as *p-value < 0.1 and **p-value < 0.05.

| From Scratch | Directly Transfer | Fine-Tune | DRL-CNN |
|---|---|---|---|
| $18.93 \pm 0.87^{**}$ | $15.64 \pm 1.25$ | $17.39 \pm 1.04^{*}$ | $15.78 \pm 1.33$ |

train from scratch for at least 3 hours, have difficulties in handling diverse urban scenarios. These experiments on transferability highlight the excellent adaptability of our model in the face of changing urban regions and OD flows, underscoring its practical utility in real-world applications.

### A.6   HYPER-PARAMETER STUDY

We tune the hyper-parameters of MetroGNN with a series of hyper-parameter studys. Specifically, we investigate two key hyper-parameters of our model in this section, which are the number of GNN layers, the dimenstion of GNN representations.

**GNN Layers.** We propose a topology-aware and OD-aware message passing mechanism that propagates through heterogeneous edges in a single GNN layer. Stacking multiple GNN layers can broaden each node's perception field, allowing it to aggregate features from distant nodes. However, excessive layer stacking can lead to oversmoothing and a decline in performance (Chen et al., 2020). We systematically vary the number of GNN layers and evaluate the variants' performance. Figure 7(a) shows that the model with 2 GNN layers achieved optimal performance, while models with more or fewer layers exhibited varying degrees of performance degradation, with an average drop of 10%.

**GNN Embedding Dimension.** The dimension of GNN embeddings closely related to their representational capacity. A higher dimension allows the model to capture more complex spatial relationships in OD flows between different regions. However, excessively high dimensions can lead to overfitting, resulting in significant performance degradation. Conversely, too low a dimension may inhibit the model's ability to learn effective representations. Meanwhile, with too low dimension, model fails to learn effective representations. We investigated the impact of varying the dimension of node embeddings on model performance. As illustrated in Figure 7(b), setting the dimension to 32 for each node resulted in the best performance. Increasing the dimension to 64 led to a performance decrease of over 10%, while reducing it to 16 and 8 resulted in performance degradations of 2.5% and 9.8%, respectively.

Table 5: The expansion performance with varying OD trips.Statistical significance is determined using a t-test to compare MetroGNN with DRL-CNN.

| Method | 1 | 2 | 3 | 4 | 5 | 6 | 7 | 8 | 9 | 10 |
|---|---|---|---|---|---|---|---|---|---|---|
| MPG | 12.71 | 12.49 | 11.87 | 9.18 | 13.54 | 15.77 | 14.53 | 18.80 | 13.61 | 14.86 |
| DRL-CNN | 15.93 | 14.31 | 15.86 | 15.10 | **16.76** | 15.71 | 16.57 | 17.29 | 16.57 | **17.98** |
| MetroGNN** | **18.63** | **16.93** | **17.91** | **16.58** | 16.69 | **19.44** | **18.83** | **20.17** | **17.79** | 17.48 |
| impr% | 14.49 | 15.48 | 11.45 | 8.93 | -0.42 | 19.19 | 12.00 | 14.28 | 6.86 | -2.86 |

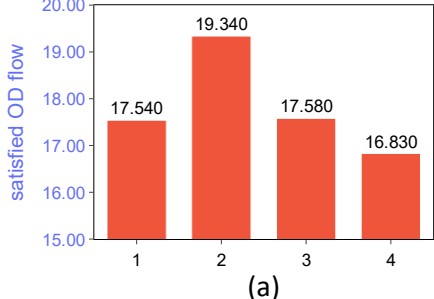 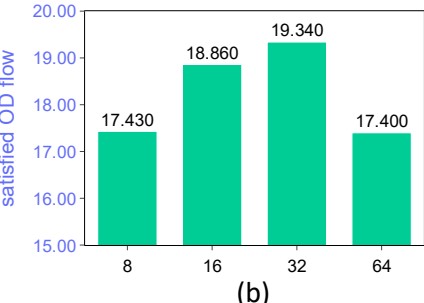

Figure 7: Performance of MetroGNN with different values of (a) GNN layers (b) node dimension for the metro network expansion in Beijing, China.

## A.7 METRO NETWORK EXPANSION WITH EQUITY

While the construction of metro networks is primarily driven by public transportation demands, social factors like equity also play a crucial role in shaping their layout (Arsenio et al., 2016; Behbahani et al., 2019). We first define the inequity of a metro network as the variance in the distance from all regions to the metro network. Let $\text{EucDis}(n_i, \mathcal{M})$ denotes to the shortest Euler distance from region $n_i$ to the metro network $\mathcal{M}$, then the inequity of $\mathcal{M}$ can be formulated as follows,

$$\text{IE}(\mathcal{M}) = \frac{1}{|\mathcal{N}|} \sum_{n_i \in \mathcal{N}} (\text{EucDis}(n_i, \mathcal{M}) - \frac{1}{|\mathcal{N}|} \sum_{n_j \in \mathcal{N}} \text{EucDis}(n_j, \mathcal{M}))^2, \quad (8)$$

and the equity improvement of the expansion is defined as the decrease in inequity $\text{IE}(\mathcal{M}_T) - \text{IE}(\mathcal{M}_0)$, where $\mathcal{M}_T$ represents the expanded metro network and $\mathcal{M}_0$ represents the original metro network. When equity is included as an evaluation metric, the metro network expansion transforms into a multi-objective optimization problem. The rewards of the MDP can be expressed as a weighted sum of two metrics as follows,

$$R_t = \alpha * (C_{od}(\mathcal{M}_t) - C_{od}(\mathcal{M}_{t-1})) + \beta * (\text{IE}(\mathcal{M}_t) - \text{IE}(\mathcal{M}_{t-1})), \quad (9)$$

where $\alpha$ and $\beta$ are the weights of the OD flows and equity, respectively. By varying the weighting factors in the rewards, we can generate expansion solutions with different preferences. As presented in Tables 6 and 8, when considering only the satisfaction of OD flows or equity individually, our method provides significantly superior expansion schemes compared to DRL-CNN, with improvements of 19.96% and 6.44%, respectively. Notably, when factoring in both travel demands and equity, MetroGNN outperforms DRL-CNN on both metrics, showcasing an average improvement of more than 20.1%. The powerful graph characterization capability of the GNN module enables the learning of intricate OD flow characteristics, while the attentive policy network correlates each region with the metro network layout, facilitating the generation of fairer expansion solutions.

Table 6: Expansion with reward weights $\alpha = 1.0, \beta = 0.0$

| Method | OD | equity | weighted |
|---|---|---|---|
| DRL-CNN | $15.78 \pm 1.33$ | $12.32 \pm 1.46$ | $15.78 \pm 1.33$ |
| MetroGNN | $18.93 \pm 0.87$** | $11.58 \pm 0.26$ | $18.93 \pm 0.87$** |
| impr% v.s. DRL-CNN | 19.96 | -6.01 | 19.96 |

Table 7: Expansion with reward weights $\alpha = 0.5, \beta = 0.5$

| Method | OD | equity | weighted |
|---|---|---|---|
| DRL-CNN | $13.20 \pm 1.72$ | $20.72 \pm 1.94$ | $16.96 \pm 1.80$ |
| MetroGNN | $16.50 \pm 1.14$** | $23.90 \pm 1.28$* | $20.20 \pm 1.21$** |
| impr% v.s. DRL-CNN | 25.00 | 15.35 | 19.10 |

Table 8: Expansion with reward weights $\alpha = 0.0, \beta = 1.0$

| Method | OD | equity | weighted |
|---|---|---|---|
| DRL-CNN | $11.86 \pm 2.32$ | $26.70 \pm 2.16$ | $26.70 \pm 2.16$ |
| MetroGNN | $10.49 \pm 1.85$ | $28.42 \pm 1.49$* | $28.42 \pm 1.49$* |
| impr% v.s. DRL-CNN | -11.55 | 6.44 | 6.44 |

