# OpenReview forum: "MetroGNN: Metro Network Expansion with Deep Reinforcement Learning"
_ICLR.cc/2024/Conference — Submitted to ICLR 2024_

### Official Review · Reviewer_m3ip · 2023-10-30

**Soundness:** 3 good
**Presentation:** 2 fair
**Contribution:** 3 good
**Rating:** 6
**Confidence:** 4

**Summary:**

In this paper, the authors investigate the problem of metro network expansion, which they formulate as an MDP. The action is selecting a region and adding it to the metro network, and the reward is the increment of satisfied OD flow. They utilize a RL framework to solve the MDP. They use a GNN network to inject spatial contiguity and OD flow into region representations. The action mask and attentive policy network are used to ensure the feasibility of the result and to reduce search space. Authors apply the method to real-world urban data and witness significant performance improvements.

**Strengths:**

S1. The notion of region is particularly interesting, as it overcomes the limitation of fixed-size grids and captures both spatial proximity and traffic flow patterns.
S2. The work addresses numerous traditional and RL-based methods as baselines, with detailed comparisons and experiments. This justifies the use of RL methods in the practical aspect. The case study of complicated scenarios also give convincing explanations.
S3. The ablation study is thorough, and strengthens the design choices of the algorithm framework.

**Weaknesses:**

W1. In the "Overall Framework" section, the MDP and RL framework do not have a formalized definition, and the description and implementation details of the overall RL framework is rather unclear.
W2. Some of the notations in the equations are not sufficiently clarified. For example, in equation (6), it is better to explicitly state that alpha_{i, j} is the relevance measure, and pinpoint the meaning of i and j.

**Questions:**

1. What is the precise definition of OD trips? How is it obtained from your dataset?
2. How do you determine whether a selected region should be an extension of an existing metro line (and which?) or the start of a new line? Is it determined in the agent's action, or by some other means?
3. In equation (5), which metro line do n_{+-1} and n_{+-2} refer to?

---

> ### Author Response · Authors · 2023-11-19
>
> Thank you very much for your positive feedback. We have added a formalized definition to both the **MDP** and the **RL framework**, and clarified some notations in the equations. All the additions are highlighted in red in the revised version, and we hope the following responses can adress your concerns.
>
> **Q1.a. What is the precise definition of OD trips?**
>
> **Response:** Specifically, if an individual starts a journey at region A, reaches region B and remians in B for at least 15 minutes, this travel event contributes a count of 1 to the OD trips from A to B. Furthermore, if the individual subsequently travels to region C, the entire movement will be considered as two separate trips: one from A to B and another from B to C. We have added this definition in Section A.2.
>
>
> **Q1.b. How is the OD trips obtained from our dataset?**
>
> **Response:** We get the spatio-temporal data from a large number of user movement trajectories from Tencent Map, a Map Service Providers in China, and the OD flow matrix in this paper is generated in the way mentioned in Q1.a. We have added the corresponding information in Experiment section.
>
> **Q2. How to determine whether a selected region should be an extension of an existing metro line (and which?) or the start of a new line?**
>
> **Response:** If the selected region **satistfies the distance and angle constraints with existing lines**, it will be considered as an extension. Specifically, if the seleted region are avaliable for multiple lines, **the station will be assigned to the closest metro line**. On the other hand, if the selected region **only satisfies the distance constraints**, it will be considered as the start of a new line. We apologize for any confusion this may have caused, and **we have updated the formulas as well as the description in Section A.1** for further clarification.
>
> **Q3. In equation (5), which metro line do $n_{\pm 1}$ and $n_{\pm 2}$ refer to?**
>
> **Response:** The **$n_{\pm 1}$ represents all the terminal stations** within the metro network, including both the originating and terminating stations of each metro line.
> The **$n_{\pm 2}$ represents all the subterminal stations** directly connected to the $n_{\pm 1}$.
> Sorry for any confusion caused by the unclear expression, and we have added the appropriate explanation in Method section. To avoid possible ambiguity, the original expression of $n_{\pm 1}$ and $n_{\pm 2}$ has been changed to $n_j \in \mathcal V_{\pm 1}$ and $n_k\in\mathcal V_{\pm 2}$  in Section A.1 of the revised manuscript.
>
> Thank you again for your constructive comments and valuable feedback, and we hope the above response can address your concerns.

---

### Official Review · Reviewer_WxNe · 2023-10-30

**Soundness:** 2 fair
**Presentation:** 2 fair
**Contribution:** 2 fair
**Rating:** 5
**Confidence:** 4

**Summary:**

This work addresses the metro network expansion problem, in which the goal is to decide which edges to add to a metro graph such as to optimize the total satisfied flow between origin-destination pairs in the network, subject to a budget constraint. The authors approach this problem by formulating it as a Markov Decision Process and proposing a reinforcement learning method that uses graph neural networks for encoding state information. More specifically, the authors consider two types of features in the GNN design: spatial features and flows, which are concatenated. The authors compare the proposed method with a variety of classic optimization algorithms and a recent RL approach, showing gains in optimality over these methods.

**Strengths:**

Originality: the work applies a method based on RL and GNNs to a new problem.  This intersection, by now, has a growing body of literature and is a fairly common "recipe".  It is moderately original in the design of its approach.

Quality: the quality of the paper is reasonable, but I have substantial concerns about the soundness of the evaluation as well as its lack of clarity in places.

Clarity: the organization of the paper is clear. The writing is of reasonable quality. The main issue in terms of clarity is the lack of precision in the description of the problem / solution method.

Significance: the work is of interest to the ICLR community, and belongs to the growing machine learning for combinatorial optimization literature.

**Weaknesses:**

**W1**. A major weakness is the fact that the authors do not report aggregated results across several runs together with statistical confidence measures (e.g. error bars). These should be presented for all figures and tables in order to account for the stochasticity of model training. Otherwise, a possible alternative explanation for the observed results is that one of the "lucky" seeds was used, which yields better performance than what we might see in the average case. It is not possible, in my opinion, to draw reliable conclusions from the presented results. In case multiple runs were indeed carried out, these details should be reported.

**W2**. Lack of (mathematical) precision in the description of the problem and solution method. The authors should aim for a level of clarity that would enable someone to reproduce the results starting from the descriptions only. This is currently not the case. Some examples where this is apparent:

- Most importantly, the MDP components (currently in Appendix A) should be formalised mathematically and not described only in plain English.
- Equation 1: presumably $i \neq j$, given self-loops are not allowed?
- Equation 3: $\mathcal{N}$ was already used to denote the entire set of nodes, whereas in this equation it is used to denote the neighbourhood. They are not the same, unless the graph is fully connected.
- The set of input features to the GNN should be fully described in the Method section and not only when discussing the ablation results (4.4).

**W3**. Potential limited scalability: the authors consider networks with up to 60 nodes, whereas the real Beijing network has approximately ~500 nodes. Hence, the problem as considered is a simplification, and this should be acknowledged. I expect that the observed performance improvement does not come "for free", and that the method suffers in scalability and has substantially longer running times than the classic methods. Studying the scalability of the method (around what number of nodes does it fail to find satisfactory solutions compared to the baseline) and adding representative runtimes for the methods would improve the manuscript.

**Questions:**

**C1**. The writing contains some important inaccuracies that should be fixed:

- "To achieve efficient search of the NP-hard problem" -> the solution space, not the problem itself, is being searched
- "The proposed model [...] successfully reduces the large solution space"; "the attentive policy network reduces the solution space drastically": as far as I can tell, the model itself does not reduce the solution space; rather, it may indeed be more efficient in how this search space is navigated.

**C2**. Typos: consider running a spellcheck. Some I have spotted: generic -> genetic (p2), maksed -> masked (p4), donate -> denote (p5)

**C3**. The analysis in 4.5 is tenuous, especially given the lack of multiple runs. For example, if we were to stop the process at step 60, we could draw a different conclusion from the one presented in this paragraph. Does this finding repeat across 10+ runs?

**C4**. The following recent paper considers metro network planning as a case study and also uses a reinforcement learning approach. In my opinion, while not directly comparable, it should be cited:

> Darvariu, V. A., Hailes, S., & Musolesi, M. (2023). Planning spatial networks with Monte Carlo tree search. Proceedings of the Royal Society A, 479(2269), 20220383.

---

> ### Author Response · Authors · 2023-11-19
>
> Thank you for your valuable feedback. We have added complete mathematical description of the MDP and RL framework in the Method section and Appendix A.5. More importantly, we provide experimental results across several runs with p-values and error bars. We sincerely hope the following responses can address your concerns.
>
> **Q1. The authors do not report aggregated results across several runs together with statistical confidence measures (e.g. error bars).**
>
> **Response:** Thank you for your suggestions regarding the reliability of our method. In the previous version, we run multiple experiments with different seeds, and only report the best results in the original manuscript. According to your valuable suggestion, in the revised version, **we have integrated the results of multiple (10) runs of the experiment in the Experiment section**, please refer to the Table 2, Figure 3 and Figure 6 in the revised manuscript.
>
> **Q2.a. The model simplifise the Beijing network with only 50 nodes.**
>
> **Response:** We are sorry for making you confused of the graph size. Actually, the graphs for Beijing and Changsha contain a total of **1,166 and 469 nodes**, respectively. It is possible that "B=50" in original Table 1 (Table 2 in revised version) caused this misunderstanding, which represents the expansion budget of $50 billion. We apologize for the ambiguity and have added detailed descriptions of the dataset in Table 1 and provided more precise descriptions.
>
>
>
> | City    | Beijing | Changsha |
> | ------- | ------- | -------- |
> |   Nodes      |   1166      |     469     |
> |   Edges      |   4656      |     1642     |
> |   Latitude       |  [39.5410,40.2319]       |    [27.9295,28.4276]      |
> | Longitude  |    [116.0385,116.8612]     |   [112.7357,113.2599]    |
>
>
>
>
> **Q2.b.  The model has substantially longer running time than classic methods.**
>
> **Response:** Thank you very much for pointing out this important issue. We would like to provide more explanations in this response.
> Firstly, **classic methods do not have a significant runtime advantage over our method**. For example, the evolutionary algorithms take an average of **2 hours**, while the mathematical programming methods take **over 3 hours and need additional manual adjustments** for the metro expansion of Beijing network.
> Secondly, those classic methods **can not generate quality and stable solution** for metro network expansion, and **must regenerated from scratch for different metro networks**.
> In contrast, though MetroGNN requires 8 hours of training, **it constantly generate optimal and stable solutions**. Moreover, the complementary experiments show that **it takes only 20 seconds** for the well-trained agent to generate better metro expansion solutions for different metro networks compared with DRL-CNN.
> According to your valuable comment, we have added the above discussion to Appendix A.5, and please check the revised manuscript for details.
>
> **Q3. The model does not reduce the solution space but enhance the efficiency of search.**
>
> **Response:** Thank you very much for the correction, we have changed the corresponding statement in the revised manuscript.
>
> **Q4. The analysis in 4.5 is tenuous, especially given the lack of multiple runs.**
>
> **Response:** Thank you for your valuable advice. We have added the results with multiple runs to the Figure 6 in the revised manuscript. The results show that the disadvantage of the attentive policy network is just a coincidence. In particular, the model with attentive policy network consistently generates better expansion solutions than the mlp-only policy network in the case of a complete 100 iterations.
>
>
> **Q5. Recent paper [2] that considers metro network planning with reinforcement learning approach should be cited.**
>
> **Response:** Thank you for your reference to this paper. We have carefully gone through the works mentioned in your comment and searched for more related literature. Despite the paper you mentioned, we also found the following two related papers [3,4] and have added proper references in Introduction and Related section:
>
> [3] Yolcu, E., & Póczos, B. (2019). Learning local search heuristics for boolean satisfiability. Advances in Neural Information Processing Systems, 32.
>
> [4] Wang, Q., & Tang, C. (2021). Deep reinforcement learning for transportation network combinatorial optimization: A survey. Knowledge-Based Systems, 233, 107526.
>
> Thank you again for your constructive comments and valuable feedback, and we hope the above response can address your concerns.

---

> > ### Comment · Reviewer_WxNe · 2023-11-21
> > **Post-rebuttal response to authors**
> >
> > Thanks for engaging with my points! Some further comments / replies:
> >
> > **W2**. I think the presentational issues and the overall lack of mathematical clarity are still outstanding (it seems the authors focused more on additional experiments). Some issues: the MDP definition is still imprecise. The GNN embeddings are definitely not part of the state itself, but are a representation based on the topology. The state is independent of how you choose to represent it for RL. The action space should be defined by specifying the constraints, and not when discussing the masking of the neural net outputs (which itself is a fairly trivial technique).  I'd suggest discussing the MDP definition in the main text before giving the details of the embedding technique.
> >
> > **Q1**. Regarding the statistical test: which test is carried out needs to be specified (e.g., t-test, Mann-Whitney U, etc.), and the authors should make sure the assumptions are satisfied. What are the significance values showing -- the results are comparing MetroGNN *and what else*? I still think 95% confidence intervals should be shown for every method (e.g., using the plusminus sign). Furthermore, Figure 5 should also include error bars.
> >
> > **Q2**. My main concern about limited scalability has been addressed well. On this point, I think the min / max lat / lon rows in Table 1 do not add anything and should be removed. If you want to keep this as a table, consider adding other informative metrics (e.g. mean degree, average / maximum path length). Otherwise, you might delete entirely and inline the description of the number of nodes and edges.
> >
> > Given the presentational and evaluation issues, in my opinion the paper is not ready for publication in its current form. Given the lack of precision in the presentation I also have a certain level of skepticism regarding the reliability of the method and the presented results. I do think that the results are promising and I would encourage the authors to resubmit to another venue after taking the time address these comments thoroughly.

---

> > > ### Author Response · Authors · 2023-11-22
> > > **Response to reviewer WxNe Part1**
> > >
> > > Thank you again for your insightful and constructive feedback, and we have further **corrected the improper mathematical description of the MDP and added results of multiple runs** according to your suggestions. We hope the following respones can address your concerns.
> > >
> > > **W2:The presentational issues and the overall lack of mathematical clarity are still outstanding.**
> > >
> > > **Response:** We acknowledge the need for clarity in our mathematical formulations, particularly in the definition of the MDP, and we are sorry for the inprecise formulation. We have revised the mathematical expressions of MDP and discussed them in the Method section of the main text with more precise descriptions:
> > >
> > > - **State:** The state $S_t$ is a three-tuple $(\mathcal{M}_t, b_t, l_t)$ containing the current metro network $\mathcal{M}_t=(\mathcal{V}_t,\mathcal{E}_t)$, the remaining budget $b_t$, and the number of new metro lines that can be built $l_t$.
> > > - **Action:** The action $A_t$ corresponds to the selection of a single node in $\mathcal{N}$. Considering that the metro network expansion includes the extension of existing lines and the construction of new lines and needs to satisfy corresponding constraints, available actions are defined as:
> > >   $A(S_t=(\mathcal{M}_t, b_t))=\{n \in \mathcal{N} | n \in (\mathcal{X}^e(\mathcal{M}_t,b_t) \cup \mathcal{X}^c(\mathcal{M}_t,b_t,l_t)) \},$
> > > where $\mathcal{X}^e(\mathcal{M}_t,b_t)$ and $\mathcal{X}^c(\mathcal{M}_t,b_t,l_t)$ represent the set of regions that are available for extension and construction based on specific constraints, respectively. The details of diverse constraints are in Section A.1.
> > >
> > > **Q1.a: Which test is carried out needs to be specified and what are the significance values showing?**
> > >
> > > **Response:** We apologize for the ambiguity and appreciate your feedback on clarity. We have specified that t-test is applied to compare the solutions of MetroGNN (ours) and DRL-CNN. The p-values indicate a significant advantage of our method compared to DRL-CNN. This clarification is now included in the caption of Table 2.

---

> > > > ### Comment · Reviewer_WxNe · 2023-11-23
> > > > **Further response to authors**
> > > >
> > > > Thanks for the further engagement -- given the improved evidence that the evaluation is robust, I am raising my score to a 5. My assessment is that the clarity of the presentation needs improving (to an extent that is not really solvable within the space of a rebuttal window), but a basic "weight of experimental evidence" bar has now been cleared in my opinion.

---

> > > ### Author Response · Authors · 2023-11-22
> > > **Response to reviewer WxNe Part2**
> > >
> > > **Q1.b: 95% confidence intervals should be shown for every method, and Figure 5 should also include error bars.**
> > >
> > > **Response:** Thank you for highlighting the importance of reliability. For each methods, we have the results of multiple runs but only reported the optima outcome in the original version. We have incorporated 95% confidence intervals into the results obtained from 10 repeated runs for each method in Table 2. The ablation experiments are also repeated and error bars are added to Figure 5.
> > > For your convenience, we list the results of Table 2 as follows,
> > >
> > > | Method          | Beijing B=40         | Beijing B=50         | Beijing B=60         | Changsha B=40        | Changsha B=50        | Changsha B=60        |
> > > | --------------- | -------------------- | -------------------- | -------------------- | -------------------- | -------------------- | -------------------- |
> > > | GS              | 8.25$_{\pm 0.00}$    | 9.31$_{\pm 0.00}$    | 10.40$_{\pm 0.00}$   | 10.11$_{\pm 0.00}$   | 11.26$_{\pm 0.00}$   | 12.58$_{\pm 0.00}$   |
> > > | GA              | 9.95$_{\pm 1.78}$    | 10.13$_{\pm 1.98}$   | 12.87$_{\pm 2.27}$   | 14.24$_{\pm 1.46}$   | 15.34$_{\pm 1.65}$   | 16.55$_{\pm 1.89}$   |
> > > | SA              | 9.59$_{\pm 1.57}$    | 10.70$_{\pm 1.59}$   | 12.29$_{\pm 2.08}$   | 13.84$_{\pm 1.63}$   | 15.02$_{\pm 1.55}$   | 16.39$_{\pm 1.72}$   |
> > > | ACO             | 11.01$_{\pm 1.14}$   | 12.42$_{\pm 1.30}$   | 13.66$_{\pm 1.45}$   | 16.61$_{\pm 1.59}$   | 17.67$_{\pm 1.52}$   | 17.17$_{\pm 1.97}$   |
> > > | MPC             | 14.40$_{\pm 0.28}$   | 15.11$_{\pm 0.61}$   | 16.60$_{\pm 1.19}$   | 17.47$_{\pm 0.60}$   | 18.34$_{\pm 0.91}$   | 20.43$_{\pm 1.30}$ |
> > > | MPG             | 14.40$_{\pm 0.28}$   | 15.16$_{\pm 0.93}$   | 16.81$_{\pm 1.13}$ | 17.47$_{\pm 0.91}$   | 18.07$_{\pm 1.26}$   | 20.17$_{\pm 1.63}$   |
> > > | DRL-CNN         | 14.46$_{\pm 0.92}$ | 15.78$_{\pm 1.33}$ | 16.38$_{\pm 1.48}$   | 18.30$_{\pm 1.03}$ | 18.98$_{\pm 1.98}$ | 19.21$_{\pm 1.81}$   |
> > > | MetroGNN (ours) | **15.88**$^{*}_{\pm 0.73}$   | **18.93**$^{**}_{\pm 0.87}$  | **21.45**$^{**}_{\pm 1.02}$ | **20.79**$^*_{\pm 1.07}$   | **22.72**$^{**}_{\pm 1.13}$  | **24.65**$^{**}_{\pm 1.36}$ |
> > > | impr\% v.s. DRL-CNN | +9.8\%            | +20.0\%           | +31.0\%           | +13.6\%           | +19.7\%           | +28.3\%           |
> > >
> > >
> > >
> > > **Q2. Consider adding other informative metrics to Table 1.**
> > >
> > > **Response:** We agree that certain metrics in Table 1 may not be directly related to the expansion task. We have added informative metrics such as Avg Degree and Avg Topological Path Length (T-APL) to Table 1,
> > > |               | Beijing | Changsha |
> > > |---------------|----------|---------|
> > > | Nodes         | 1166     | 469     |
> > > | Edges         | 4656     | 1642    |
> > > | Avg Degree    | 3.99     | 3.50    |
> > > | Avg Area      | 1.21 km² | 1.17 km²|
> > > | T-APL         | 20.37    | 13.02   |
> > > | T-MSPL        | 55       | 39      |
> > > | M-APL         | 34.38 km | 20.81 km|
> > > | M-MSPL        | 101.73 km| 63.48 km|
> > >
> > > where **T** denotes topological measures, while **M** denotes metric measures. **APL** represents average path length, and **MSPL** refers to maximum shortest path length.
> > >
> > > We sincerely appreciate your thorough review and hope these updates address your concerns. Should you have any further questions or require additional clarification, please feel free to let us know, Wwe are more than willing to provide further clarification on any remaining points.

---

### Official Review · Reviewer_6Gv3 · 2023-10-31

**Soundness:** 3 good
**Presentation:** 3 good
**Contribution:** 2 fair
**Rating:** 6
**Confidence:** 4

**Summary:**

This paper proposes a graph-based Reinforcement Learning (RL) framework to solve the metro network expansion task (a geometrical combinatorial optimization problem) for maximizing overall OD flow satisfaction with several constraints, e.g., total budget, spacing between stations, and line straightness. The proposed framework, MetroGNN, incorporates Graph Neural Networks (GNN) and an attentive policy network with an action mask to learn representations for urban regions and select new metro stations. The experiments conducted on real-world urban data of Beijing and Changsha demonstrate that the proposed MetroGNN can improve OD flow satisfaction by over 30% against the state-of-the-art RL-based approach.

**Strengths:**

1. This paper proposes to solve a complex metro network expansion problem by using a graph-based reinforcement learning framework. The problem is significant, and the solution makes sense to me.
2. The proposed approach is evaluated on two real-world urban datasets collected from two Chinese metropolises, Beijing and Changsha, which demonstrates its effectiveness in improving the overall OD flow satisfaction.
3. This paper is overall well-written and easy to follow. The illustrations are clear and helpful to understand this paper.

**Weaknesses:**

1. The technical contributions of this work are limited. While the metro network expansion task is essentially a transportation network combinatorial optimization problem, there have been many existing works [1] studying how to apply RL combined with GNN or attention to address it. It seems the authors only introduce some of the same or similar methods to a specific combinatorial optimization problem. However, there are no substantial technical innovations.
2. While this work investigates a realistic metro network expansion problem, it only aims to optimize the total satisfied OD flow. However, many other factors need to be considered and optimized to construct a realistic metro network, e.g., social equity or fairness, environmental impact, and revenues. It’s hard to evaluate whether the proposed method is applicable in real scenarios.
3. Some important experimental setups are not mentioned or clearly described. For example, the statistics and analysis of datasets. The implementation and hyper-parameter details of baselines. Such information is very significant for the evaluation of experimental reliability.

[1] Wang Q, Tang C. Deep reinforcement learning for transportation network combinatorial optimization: A survey[J]. Knowledge-Based Systems, 2021, 233: 107526.

**Questions:**

1. The collected real OD flow data are based on the realistic transportation network. If the metro networks have be changed, how can the authors obtain the corresponding OD flow?

---

> ### Author Response · Authors · 2023-11-19
>
> Thank you for recognizing our work. We sincerely hope the following responses can address your remaining concerns.
>
> **Q1. The authors only introduce some of the same or similar methods to a specific combinatorial optimization problem without substantial technical innovations.**
>
> **Response:** Thank you for your comment. We would like to summarize the innovations of our work as follows:
>
> - **We propose a specifically designed GNN model for heterogeneous multi-graphs.** This model enables effective capturing of intricate patterns in OD trips which is overlooked by traditional methods.
> - **We scale up the metro network more than tenfold compared to existing studies.** The attentive policy network and carefully designed action mask enhances the agent's efficiency in exploring the solution space, enabling our proposed RL model to handle urban region graphs of more than one thousand nodes.
> - **Our proposed model exhibits remarkable transferability.** The trained model demonstrates its ability to quickly generate high-quality expansion solutions when faced with different metro networks or varying OD trips. Please refer the response to Q2 of reviewer zGme for more details. This adaptability distinguishes our approach and enhances its applicability across diverse scenarios.
>
>
> **Q2. Many other factors need to be considered and optimized to construct a realistic metro network, e.g., social equity or fairness, environmental impact, and revenues.**
>
> **Response:** We genuinely appreciate your comment on the problem formulation and completely agree that metro network expansion requires careful consideration of multiple factors. On the one hand, the primary objective of the metro network is to effectively address the travel needs of citizens, thus we believe that the OD flow is a decisive factor in the planning of the metro network. On the other hand, we add additional experiments that incorporate the **social equity** consideration. Specifically, we quantify the inequity as the **population-weighted variance of the shortest distance from regions to the metro network**, and use the decline in inequality for evaluation. We also introduce $\alpha$ and $\beta$ for the adjustment of the weighting between OD flows and equity, respectively. As shown in the following table (higher is better), **our proposed MetroGNN consistently outperforms DRL-CNN under different weightings, achieving better expansion targets with lower variance** (\* denotes to p-value < 0.1, and \*\* denotes to p-value < 0.05).
>
>
> |       Method        | $\alpha=1.0,\beta=0.0$ |                 |                     | $\alpha=0.5,\beta=0.5$ |                 |                     | $\alpha=0.0,\beta=1.0$ |                 |                    |
> |:-------------------:| ---------------------- | --------------- | ------------------- | ---------------------- | --------------- | ------------------- | ---------------------- | --------------- | ------------------ |
> |                     | OD                     | equity          | weighted            | OD                     | equity          | weighted            | OD                     | equity          | weighted           |
> |       DRL-CNN       | 15.78$\pm$ 1.33        | 12.32$\pm$ 1.46 | 15.78$\pm$ 1.33     | 13.20$\pm$ 1.72        | 20.72$\pm$ 1.94 | 16.96$\pm$ 1.80     | 11.86$\pm$ 2.32        | 26.70$\pm$ 2.16| 26.70$\pm$ 2.16|
> |MetroGNN| 18.93$\pm$ 0.87  ** |   11.58$\pm$ 0.26     | 18.93$\pm$ 0.87  ** |      16.50$\pm$ 1.14 **                 |   23.90$\pm$ 1.28     | 20.20$\pm$ 1.21  ** |                 10.49$\pm$ 1.85       |  28.42$\pm$ 1.49  *      | 28.42$\pm$ 1.49  * |
> | impr% v.s.  DRL-CNN |            19.96            |   -6.01     | 19.96               |           25.00             |    15.35    | 19.10               |             -11.55           |    6.44    | 6.44               |
>
>
> Above results have been added in Section A.7.

---

> ### Author Response · Authors · 2023-11-19
>
> **Q3. Some important experimental setups are not mentioned or clearly described. For example, the statistics and analysis of datasets and implementation details of baselines.**
>
> **Response:** Thank you for bringing this to our attention. We have added the details of the dataset at Experiment section, and the implementation of baselines are added at Appendix A.3.
>
> **Q4. The collected real OD flow data are based on the realistic transportation network. If the metro networks have been changed, how can the authors obtain the corresponding OD flow?**
>
> **Response:** Thank you for this valuable comment. Though there are actually no methods that can predict the OD flow after expansion, research [1] provides an idea for evaluating the feasibility of a metro network to cope with changing OD flows. Following [1], we assume that OD flows do not change drastically after the expansion of the metro network, and add a 5% random noise to all OD trips. As shown in the follow table, **MetroGNN outperforms the DRL-CNN with p-value < 0.003** and allows small variations in OD flow (experiments are repeated for 10 times with different random seeds).
>
>
>
> | Method              | 1         | 2         | 3         | 4         | 5         | 6         | 7         | 8         | 9         | 10        |
> | ------------------- | --------- | --------- | --------- | --------- | --------- | --------- | --------- | --------- | --------- | --------- |
> | MPG                 | 12.71     | 12.49     | 11.87     | 9.18      | 13.54     | 15.77     | 14.53     | 18.80     | 13.61     | 14.86     |
> | DRL-CNN             | 15.93     | 14.31     | 15.86     | 15.10     | **16.76** | 15.71     | 16.57     | 17.29     | 16.57     | **17.98** |
> | MetroGNN**          | **18.63** | **16.93** | **17.91** | **16.58** | 16.69     | **19.44** | **18.83** | **20.17** | **17.79** | 17.48     |
> | impr% v.s.  DRL-CNN | 16.95     | 18.31     | 12.93     | 9.80      | -0.42     | 23.74     | 13.64     | 16.66     | 7.36      | -2.78     |
>
>
> [1] Wang, L., Jin, J. G., Sibul, G., & Wei, Y. (2023). Designing Metro Network Expansion: Deterministic and Robust Optimization Models. Networks and Spatial Economics, 23(1), 317-347.
>
> The above results are shown in Section A.5 of the revised manuscript.
>
> Thank you again for your constructive comments and valuable feedback, and we hope the above response can address your concerns.

---

> ### Author Response · Authors · 2023-11-21
> **Willing to further clarify your remaining concerns**
>
> Dear reviewer 6Gv3,
>
> We greatly appreciate your thorough review and positive feedback. In response to your comments, we have clarified the contributions of our work and conducted additional experiments, including the incorporation of societal equity metrics and robustness testing under varying OD trips. We would be grateful if you could confirm whether our responses have effectively addressed your concerns. We are more than willing to provide further clarification on any remaining points.
>
> Thank you for your time and consideration!

---

> > ### Comment · Reviewer_6Gv3 · 2023-11-22
> >
> > I thank the authors for their detailed responses to my comments. However, I still have concerns regarding W1. Despite this is an application paper, the GNN encoder is a straightforward adaption. Regarding scalability and transferability, if those are the key innovations of this paper, the authors are suggested to elaborate more about the design insight of the framework about which part the scalability and transferability come from and why they outperform existing baselines.

---

### Official Review · Reviewer_zGme · 2023-11-01

**Soundness:** 3 good
**Presentation:** 1 poor
**Contribution:** 2 fair
**Rating:** 3
**Confidence:** 3

**Summary:**

Selecting urban regions for traffic route construction to maximize origin-destination flow is a hard optimization problem because the solution space grows exponentially on the number of nodes. This paper models this problem as a MDP and applies reinforcement learning (RL) algorithms to search for a good solution.  This paper uses a graph neural network to learn the state representation and use action masks to rule out unavailable actions. The empirical results show this method increases the total origin-destination flow by 30% compared with state-of-the-art methods.

**Strengths:**

1. This paper builds an end-to-end reinforcement learning algorithm to find a good solution in a combinatorial optimization problem - traffic routes construction. To learn this solution efficiently, this paper builds a graph neural network to learn the state representation.

2. Its empirical results show this method increases the total origin-destination flow by 30% compared with state-of-the-art methods.

**Weaknesses:**

1. My major concern is that this paper does not have enough novelty to be published in a top machine learning venue. Indeed, selecting urban regions for traffic route construction to maximize origin-destination flow is a hard optimization problem because the solution space grows exponentially on the number of nodes. However, reinforcement learning (RL) has been known to be a useful tool for searching solutions in a large solution space since 1996 [1].

2. Moreover, using graph neural networks to learn state representation is also not a novel technique. The network proposed in this paper is not well-justified to have sufficient novelty.

3. Using action masks to eliminate infeasible regions is also a common approach in RL applications.

4. Other than insufficient novelty in the algorithm, the model built by this paper is also preliminary. For example, it is natural to consider that more regions could emerge as the city is expanding. The model in this paper is obviously not a high-fidelity model that could be used in real construction.

5. This paper is not well-written. The details of the model and algorithms are not defined in a clear and mathematical way. The presentation and coherence of this paper could be greatly improved by deleting excessive words and sentences.

[1] Bertsekas, D. P. and Tsitsiklis, J. N. (1996). Neuro-Dynamic Programming. Athena Scientific Belmont, MA.

**Questions:**

N/A

---

> ### Author Response · Authors · 2023-11-19
> **Response to reviewer zGme Part1**
>
> Thank you for your valuable comments and we sincerely hope the following responses can address your concerns.
>
> **Q1. This paper does not have enough novelty to be published in a top machine learning venue. The reinforcement learning, GNNs and action masks are already widely used.**
>
> **Response:** Thank you for your concern on the innovation of our work. We would like to summarize the three major technical innovations of our work as follows:
> - **We propose a novel GNN model specially designed for heterogeneous multi-graphs.** It allows us to effectively capture the intricate patterns of OD trips, while traditional methods merely focus on the spatial information of regions.
> - **We scale up the problem to more than ten times larger than those considered in previous studies.** With the attentive policy network and the meticulously designed action mask, the agent can search the solution space much more efficiently, enabling our proposed RL model to handle urban region graphs of more than one thousand nodes.
> - **Our proposed model exhibits excellent transferability.** The trained model demonstrates the ability to swiftly generate high-quality expansion solutions when confronted with different metro networks or varying OD trips. Please refer the response to Q2 for more details. This adaptability sets our approach apart and enhances its utility across diverse scenarios.
>
>
>
> **Q2. The model built by this paper is also preliminary, not considering cities' expansion.**
>
> **Response:** Thank you for your constructive comment on the expansion of cities, and we have added a series of experiments to showcase how our model handle the growing urban regions. Firstly, **we remove 20% of the fringe regions of the city to train our model and then applied the trained model to the original complete urban regions**. The following table illustrate the strong transferability of MetroGNN. When directly applied to a larger city, MetroGNN can quickly (within 20 seconds) generate solutions which are comparable to DRL-CNN. **Notably, the fine-tuned model shows an improvement of more than 10.2% over the DRL-CNN** (\* denotes to p-value < 0.1, and \*\* denotes to p-value < 0.05).
> | from scratch**  | directly transfer | fine-tune*      | DRL-CNN         |
> | --------------- | ------------------ | --------------- | --------------- |
> | 18.93 $\pm$ 0.87 | 15.64 $\pm$ 1.25 |  17.39 $\pm$ 1.04   | 15.78 $\pm$ 1.33 |
>
> Secondly, **we introduce 5% noise to OD trips data with 10 different seeds to simulate the changes with urban development**. The following table demonstrates that **MetroGNN can allow variations in OD flow and still outperforms existing approaches**.
>
> | Method              | 1         | 2         | 3         | 4         | 5         | 6         | 7         | 8         | 9         | 10        |
> | ------------------- | --------- | --------- | --------- | --------- | --------- | --------- | --------- | --------- | --------- | --------- |
> | MPG                 | 12.71     | 12.49     | 11.87     | 9.18      | 13.54     | 15.77     | 14.53     | 18.80     | 13.61     | 14.86     |
> | DRL-CNN             | 15.93     | 14.31     | 15.86     | 15.10     | **16.76** | 15.71     | 16.57     | 17.29     | 16.57     | **17.98** |
> | MetroGNN**          | **18.63** | **16.93** | **17.91** | **16.58** | 16.69     | **19.44** | **18.83** | **20.17** | **17.79** | 17.48     |
> | impr% v.s.  DRL-CNN | 14.49     | 15.48     | 11.45     | 8.93      | -0.42     | 19.19     | 12.00     | 14.28     | 6.86      | -2.86     |

---

> ### Author Response · Authors · 2023-11-22
> **Response to reviewer zGme Part2**
>
> **Q3.  The details of the model and algorithms are not defined in a clear and mathematical way, and this paper could be greatly improved by deleting excessive words and sentences.**
>
> **Response:** Thank you for your valuable comments on the presentation. We have added clear mathematical description of the MDP and the RL framework in Method section and Appendix A.1. We have also removed unnecessary expressions to make the article more concise. We provide critical descriptions of MDP as follows:
>
> - **State:** The state $S_t$ is a three-tuple $(\mathcal{M}_t, b_t, l_t)$ containing the current metro network $\mathcal{M}_t=(\mathcal{V}_t,\mathcal{E}_t)$, the remaining budget $b_t$, and the number of new metro lines that can be built $l_t$.
> - **Action:** The action $A_t$ corresponds to the selection of a single node in $\mathcal{N}$. Considering that the metro network expansion includes the extension of existing lines and the construction of new lines and needs to satisfy corresponding constraints, available actions are defined as:
>   $A(S_t=(\mathcal{M}_t, b_t))=\{n \in \mathcal{N} | n \in (\mathcal{X}^e(\mathcal{M}_t,b_t) \cup \mathcal{X}^c(\mathcal{M}_t,b_t,l_t)) \},$
> where $\mathcal{X}^e(\mathcal{M}_t,b_t)$ and $\mathcal{X}^c(\mathcal{M}_t,b_t,l_t)$ represent the set of regions that are available for extension and construction based on specific constraints, respectively. The details of diverse constraints are in Section A.1.
> - **Reward:** Referring to the definition of total satisfied OD flow in Equation 1, the intermediate reward $R_t$ for action $A_t$ is defined as ${C_{od}(\mathcal M_t)-C_{od}(\mathcal{M}_{t-1})}$.

---

> ### Author Response · Authors · 2023-11-23
> **Looking forward to further discussion**
>
> Dear reviewer zGme,
>
> In our response, we have provided clear illustrations of our contributions, enhanced the mathematical description of our model, and conducted supplementary experiments according to your constructive suggestions. We would greatly appreciate it if you could take a moment to confirm whether our responses have effectively addressed your concerns. We are looking forward to any additional feedback you may have.
>
> Thank you for your time and consideration!

---

### Author Response · Authors · 2023-11-19
**Response to all reviewers**

We express our sincere gratitude to all the reviewers for providing valuable feedback on our work. Based on the reviews from all reviewers, we provide the following clarifications with extensive supplementary experiments:

1. **The contribution of our work can be summarized into three major technical innovations.** We take into account all the reviewers' comments and suggestions and explain the innovation of our work as follows: Firstly, we introduce a novel GNN model specifically tailored for heterogeneous multi-graphs, effectively capturing intricate patterns within OD trips. Secondly, the incorporation of the attentive policy network and the meticulously designed action mask allow us to scale the problem to more than ten times larger than previous studies. Thirdly, our proposed model exhibits exceptional transferability, consistently generating high-quality and stable expansion solutions within 20 seconds, even for dynamic scenarios involving changing cities and OD flows.
2. **We add clear descriptions of the dataset and algorithm.** We provide more mathematical details regarding our proposed model and algorithm, as suggested by reviewer zGme and WxNe. Additionally, we have included comprehensive statistics on the dataset, implementation details of baselines, and a complete definition of OD trips, responding to the constructive feedback from reviewers 6Gv3 and m3ip. Cases of ambiguous or mixed use of symbols and the misrepresentation of the solution space have been corrected , as suggested by reviewer WxNe and m3ip.
4. **We add more experiments to showcase the effectiveness and robustness of our approach.** We have updated the results with multiple runs and added  error bars and variance measures in tables and figures, as suggested by reviewer WxNe. We apply our model to different metro network expansion scenarios, such as changing urban area and OD trips, as suggested by reviewer zGme and 6Gv3. Expansion with other considerations like equity is also investigated, as suggested by reviewer 6Gv3. These experiments collectively demonstrate that MetroGNN can adeptly adapt to varying scenarios and different evaluation metrics.

Our work addresses a challenging geometrical CO problem of metro network expansion. We sincerely hope that our response can address your concerns. We are more than willing to engage in further discussion if you have any questions.

---

### Meta-Review · Area_Chair_BG12 · 2023-12-08

**Metareview:**

(a) Summarize the scientific claims and findings of the paper based on your own reading and characterizations from the reviewers.
- The paper designs an RL agent for the combinatorial problem of metro-network expansion.

(b) What are the strengths of the paper?
- The paper proposes an RL solution to a difficult combinatorial optimization problem. This seems valuable since the original problem is difficult even for existing opt. solvers and current heuristics (this is in contrast to using RL to solve something like TSP problems for which we already have good opt. solvers).
- The proposed method outperforms current ML/RL approaches

(c) What are the weaknesses of the paper? What might be missing in the submission?
- The proposed method is known
- Even after the revision, there were still remarks that the clarity of the manuscript should be improved
- The setting is still semi-synthetic and so the real-world impact of the results is unknown (especially compared to existing baselines)
- It is unclear if the work generalizes beyond this particular problem

**Justification For Why Not Higher Score:**

The application domain (metro network expansion) is the main interest of this work. The machine learning techniques, by all accounts, are fairly standard, even if there are certain elements of novelty. It also seems like these elements could/should be better emphasized (through discussions and analyses for example) in the manuscript.

The results show that the approach scales up and can be transferred to varying OD. This is certainly interesting, but these findings also align with similar work leveraging GNNs for combinatorial optimization and transportation problems (notably the control of traffic signals). In that sense, these findings tend to confirm current knowledge. It is also unclear if any of these findings generalize to other similar problems (in CO and/or in transportation).

The results show statistically significant gains compared to competitors (notably DRL-CNN), but it was unclear how significant these might be in practice. Overall, it would be useful to discuss the sim to the real gap further (in other words, give an idea of the usefulness of the reported improvements in practice). I can imagine this to be challenging to do and so this point is more minor.

Part of the ICLR audience is interested in work at the intersection of ML and combinatorial optimization. However, this paper seems to deal exclusively with a specific problem (metro extension), so it's difficult to evaluate the possible interests of the community. It is likely this work could have more impact in a transportation venue.

Overall, I find that, taken together, these weaknesses indicate the paper might have the potential to be published in a major machine-learning conference (such as ICLR), but the current manuscript requires more work. I am sorry that I cannot recommend acceptance at this stage.

**Justification For Why Not Lower Score:**

N/A

---

### Decision · Program_Chairs · 2024-01-16

Reject